# Tailoring Cod Gelatin Structure and Physical Properties with Acid and Alkaline Extraction

**DOI:** 10.3390/polym11101724

**Published:** 2019-10-21

**Authors:** Svetlana R. Derkach, Yuliya A. Kuchina, Andrey V. Baryshnikov, Daria S. Kolotova, Nikolay G. Voron’ko

**Affiliations:** 1Department of Chemistry, Murmansk State Technical University, Sportivnaya str., 13, 183010 Murmansk, Russia; uak2008@mail.ru (Y.A.K.); kolotovads@gmail.com (D.S.K.); voronkonikolay@mail.ru (N.G.V.); 2Laboratory of Biochemistry and Technology, Polar branch of Russian Federal Research Institute of Fisheries and Oceanography, Academician Knipovich str., 6, 183038 Murmansk, Russia; baryshnikov@pinro.ru

**Keywords:** cod gelatin, extraction, secondary structure, functional properties

## Abstract

Gelatin (G) was extracted from the skin of Atlantic cod at different pH of the aqueous phase (pH 3, 4, 5, 8 and 9) and at a temperature of 50 ± 1 °C. The yield of gelatin (G3, G4, G5, G8, and G9, respectively) was 49–55% of the dry raw material. The influence of extraction pH on the physicochemical and functional properties of gelatin was studied. Sample G5 was characterized by higher protein content (92.8%) while lower protein content was obtained for sample G3 (86.5%) extracted under more aggressive conditions. Analysis of the molecular weight distribution showed the presence of α- and β-chains as major components; the molecular weight of the samples ranged between 130 and 150 kDa, with sample G5 having the highest molecular weight. IR spectra of all samples had absorption bands characteristic of fish gelatin. The study of the secondary structure demonstrated higher amounts of ordered triple collagen-like helices for G5 extracted under mild conditions. Accordingly, sample G5 formed gels with high values for the storage modulus and gelling and melting temperatures, which decrease as pH changes into acidic or alkaline regions. In addition, the differential scanning calorimetry data showed that G5 had a higher glass transition temperature and melting enthalpy. Thus, cod skin is an excellent source of gelatin with the necessary physicochemical and functional properties, depending on the appropriate choice of aqueous phase pH for the extraction.

## 1. Introduction

Gelatin is a natural biopolymer. Due to its ability to form thermally reversible structures, gelatin is widely used in the food and pharmaceutical industries and in medicine [1,2]. Gelatin is a product of thermal acidic, alkaline or enzymatic destruction of collagen—a fibrillar protein present in the skin, connective tissues, bones and other organs of mammals and fish. Collagen fibrils are composed of rod-like molecules (tropocollagen) the length of which is 300 nm with thickness of 1.5 nm [3]. Tropocollagen is composed of three polypeptide chains, the so-called α-chains, which form a triple helix. The triple helix of tropocollagen is stabilized by covalent cross-links, located at the ends of polypeptide chains, the so-called telopeptides. During thermal denaturation, the covalent cross-links are destroyed, and the triple helix of the collagen macromolecule unwinds, forming free polypeptide α-chains, i.e., gelatin. Due to its dense spiral structure, the native tropocollagen is resistant to alkaline, acidic and enzymatic hydrolysis. Nevertheless, the peptide bonds can partially break to form fragments of α-chains when gelatin is obtained during long-term heat treatment.

Obtaining gelatin from collagen is a multistage process that consists of preparation of the collagen-containing raw material, extraction, purification, and drying of the aqueous extracts. The main stage of the gelatin technology is aqueous extraction at a temperature of 40–80 °C with the use of acids (A-type gelatin) and alkalis (B-type gelatin) [4,5]. When the gelatin is obtained, the higher-order collagen structures are destroyed: fibrils, microfibrils, triple helices, and other supramolecular formations. At the same time, the primary structure (amino acid sequence) of α-chains is retained, although their fragmentation is possible as a result of partial hydrolysis [3,6]. Gelatin retains its unique capacity for the limited renaturation of the collagen-like triple helices under certain conditions, which can be accompanied by gel formation.

The degree of destruction of native collagen depends on the technology of gelatin extraction and on the type of raw material containing collagen [2,7,8]. Therefore, unfractionated gelatin is a mixture of polypeptide/polymer chains with different molecular weights including α-chains (~100 kDa), β-chains (~200 kDa), γ-chains (~300 kDa) and their fragments [6]. α-Chains are intact polypeptide chains of collagen, while β- and γ-chains are covalently cross-linked double and triple compositions of α-chains, respectively.

Currently, commercial gelatin mainly comes from mammalian sources [9,10]. However, it should be noted that alternative non-mammalian species have grown in importance [11,12]. In fish raw material, collagen can be found mainly in the connective tissues and the skin, which is 80–90% composed of collagen [2]. Currently, most of the fish caught is processed for skinless fillet. Therefore, the main source of collagen-containing raw materials is waste (skin, bones, fins, etc.) from processed commercial fish, which can be effectively used.

In recent years, a considerable number of papers on the study of the properties of fish gelatin have been published [4,13], which is comparable with those on gelatin from mammalian sources [14]. The physicochemical properties of gelatin obtained from the skin of shark [15], rainbow trout [16], catfish [17], pollock [18] and cobia [19] have been studied. The properties of gelatin largely depend on the type of raw material and the conditions of aqueous extraction (temperature, processing time, pH of the reaction medium, etc.) [20,21,22]. 

One of the decisive factors in developing fish gelatin technology is the amino acid composition of the raw material (its uniqueness), which largely depends on the fish habitat. For example, the gelatin obtained from cold-water fish (cod, haddock, salmon, and pollock) [12,23] contains less of such amino acids like proline and hydroxyproline and it has a lower molecular weight in comparison with that of gelatin obtained from warm-water fish (tuna, rainbow trout and eel) [16,20,24]. As a result, gelatin from cold-water fish has inferior thermal and rheological properties of gelatin to that extracted from warm-water fish. 

Nevertheless, an increased catch of cold-water fish (fish from northern seas) makes them a promising raw material for obtaining gelatin. In the production of food gelatin, it is necessary to completely remove lipids from raw materials, which greatly complicates the technological process [25]. Therefore, to obtain high-quality gelatins, it is advisable to use the skin of fish with a low fat content, such as cod, haddock, pollock, hake, etc. In addition, it is possible to improve the thermal and rheological properties of gelatin derived from cold-water fish by modifying gelatin, for example, using complex-forming ionic polysaccharides [26].

The Atlantic cod (*Gadus morhua*) is a valuable commercial species due to a widespread habitat and its nutritious properties. Cod is characterized by a high content of essential amino acids; it has a balanced protein and lipid composition. Every year, up to 1300 thousand tons of cod are caught in the world, and a significant part of the catch is sent for processing in order to obtain skinned fillets, which leads to the accumulation of a large amount of collagen-containing by-products, i.e., skin. Therefore, development of efficient methods of cod skin processing to obtain gelatin with the physicochemical and functional characteristics appropriate for food applications is an urgent task. At the same time, a high product yield is a necessary requirement. 

Most methods of obtaining fish gelatin (including those from cod skin) consist of extraction for 8 to 20 h at an average temperature of 40–50 °C and neutral pH [4,12]. Nevertheless, considering that the properties of natural gelatins, such as charge and conformation of macromolecules, solubility, are strongly dependent on pH, it can be expected that pH value is a major determinant of gelatin extraction. Extraction done in an acidic or alkaline environment can reduce the extraction time, consequently decreasing the cost of the final product. The aim of this study was to develop a method for extracting gelatin from the skin of Atlantic cod at low (≤5) and high (≥8) pH values of the aqueous phase used for extraction. A special stress was put on how the pH affects the physicochemical and functional properties of the fish gelatin samples obtained.

## 2. Materials and Methods

### 2.1. Materials

The skin of Atlantic cod (G. morhua) was used for gelatin extraction. Atlantic cod was caught by Trawl Fleet Co., Ltd. (Murmansk, Russia) in the Barents Sea and delivered chilled to the port of Murmansk. Upon arrival to the Department of Chemistry, Murmansk State Technical University medium-sized fishes (70–90 cm long) were skinned; the skins were cleaned of muscle fragments, washed in running water, frozen and stored at −20 °C until the experiments. Gelatin “Sigma” (Sigma-Aldrich Oakville, Ontario, Canada) from cold-water fish skin was used as a control sample. All chemical reagents used in the work had analytical-grade purity (Pro Analysis). 

### 2.2. Cod Gelatin Extraction

Cod gelatin (G) was extracted from the skin of Atlantic cod following the standard procedure. The cod skin was defrosted, cut into square pieces of 5 × 5 mm and then defatted by washing with ethanol.

The cut cod skin was mixed with distilled water at a ratio of 1:3 (*w*/*w*) and stirred for 10 min. Gelatin extraction was carried out at different pH values of the aqueous phase (pH 3.0, 4.0, 5.0, 8.0 and 9.0) for 3 h at a temperature of 50 ± 1 °C with constant stirring at a speed of 80–100 rpm. The samples of gelatin (G3, G4, G5, G8, and G9, respectively) were obtained. Glacial CH_3_COOH and 4 M NaOH were used to adjust the pH of the aqueous phase.

After extraction, the reaction mixture was neutralized to pH 5.5–6.0 and then filtered. 

The method of vacuum filtration at a temperature of 30 °C was used, and a paper filter (Ekros, St. Petersburg, Russia) with a pore diameter of 12 μm (Akros, Russia) was used. The filtrate (gelatin solution) was dried in a FreeZone freeze dryer (Labconco, Kansas City, USA) at a temperature of −50 °C and a residual pressure of 3.0 Pa. The gelatin obtained was stored at 5 °C until further use.

The product yield (B, %) was calculated using the following formula:B = (m/M) × 100%,(1)
where *m* is the mass (g) of dried gelatin, and *M* is the mass (g) of dried raw material with a moisture content of up to 10%.

### 2.3. Chemical Composition of Fish Gelatin

The chemical composition of the gelatin samples was determined by standard methods (AOAC, 2016). The moisture content was determined after they had been dried to constant weight at 105 ± 5 °C; the fat content was determined by the Soxhlet method; the protein content was determined by the Kjeldahl method; amine nitrogen was determined by the formol titration method; mineral substances were determined by the method of burning samples in a muffle furnace at 550 ± 10 °C.

### 2.4. Isoelectric Point

The isoelectric point, pI, of the gelatin was determined by a capillary viscometer method using a VPZH-2 glass viscometer (Ekros, St. Petersburg, Russia) (diameter of 0.34 mm) at 25.0 ± 0.1 °C and by a turbidimetric method using a T70 UV/Visible spectrophotometer (PG Instruments, Midlands, UK) at 23.0 ± 0.5 °C.

### 2.5. Amino Acid Composition

The amino acid composition of gelatin was determined by high-performance liquid chromatography (HPLC) [27].

A Shimadzu LC-10A liquid chromatograph (Shimadzu, Kyoto, Japan) with a SUPELCOSIL LC-18 column (4.0 mm × 30 cm, 5 μm) (Sigma-Aldrich, Munich, Germany) was used. Gradient separation was carried out with binary eluent (acetonitrile/0.05 M sodium acetate solution), with an eluent flow rate of 1.5 mL/min and column temperature of 35 °C. The peaks were recorded by a spectrofluorimetric detector (RF-10 A_XL_) with an excitation wavelength of 350 nm and an emission wavelength of 450 nm.

The content of proline and hydroxyproline was determined by a LCMS-QP800a chromatograph mass spectrometer (Shimadzu, Kyoto, Japan) with an atmospheric pressure chemical ionization (APCI) module in positive ionization mode, scanning in the range of *m*/*z* = 50–400 (parameter detection *m*/*z* = 116. 1 and 132.1 M + H^+^) [28]. Separation was done using a SUPELCOSIL LC-18 column (4.0 mm × 25 cm, 5 μm) with acetonitrile solution (volume ratio of water to acetonitrile 85:15) with the addition of a 0.01 M formic acid solution; the flow rate (eluent) was 0.75 mL/min, analysis time was 18 min and sample volume was 10 μL (the gelatin was previously dissolved in a 0.05 M solution of acetic acid) [29].

To calibrate the column, standard amino acid samples (Sigma-Aldrich, Munich, Germany) were used.

### 2.6. Molecular Mass

Molecular weight distribution of gelatin samples was determined by an HPLC method. Peaks were recorded by an LCMS-QP8000 chromatography mass spectrometer (Shimadzu, Kyoto, Japan) with a spectrophotometric detector (SPD; 10 AV_VP_) at λ = 280 nm. Chromatographic separation was performed using a Tosoh TSKgel Alpha-4000 column at 25 °C in isocratic mode, a flow rate of 0.8 mL/min; the eluent was 0.15 M NaCl. For column calibration, proteins (Sigma-Aldrich, Munich, Germany) with a known molecular weight in the range of 12.4 kDa (cytochrome C) to 200 kDa (b-amylase) from were used. The mass-average molecular weight (*M*_w_, kDa) was calculated by the following equation:lg *M*_w_ = −0.001 *V* + 5.751,(2)
where *V* is the volume of the eluent flowed through the column (cm^3^).

The molecular weight composition of gelatin was determined by the method of horizontal electrophoresis in a gradient polyacrylamide gel in the presence of sodium dodecyl sulfate (PAGE) [30].

Electropherograms were obtained using a Multiphor II unit (LKB-Pharmacia, Sweden); standard polyacrylamide plates, an ExcelGel^TM^ SDS gradient, with the cast gradient, where the gel concentration was 8% to 18% (GE Healthcare Bio-Sciences). The gelatin solution (0.5–1.0 wt.%) was mixed with the buffer solution at a ratio of 1:1, kept at a temperature of 95 °C for 3 min and placed on a polyacrylamide plate. Electrophoretic separation of protein fractions was done at a current of 25 mA and a temperature of 15 °C. Then, the gel plate was stained by the Coomassie method at a temperature of 50 °C for 20 min in a 10% CH_3_COOH with 1.25 g/L Phast Gel Blue. Calibration curves based on relatively mobile standard markers, i.e., proteins (GE Healthcare, Little Chalfont, UK) with a molecular weight from 14.4 kDa (α-lactalbumin) to 97 kDa (phosphorylase b) and *β*-*amylase* (Sigma-Aldrich, Munich, Germany) with *M*_w_ = 200 kDa, were used to determine the molecular weight of the protein fractions.

### 2.7. Secondary Structure of Gelatin

FTIR spectroscopy was used to determine the secondary structure of macromolecules. FTIR spectra were obtained using an IRTracer-100 FTIR spectrometer (Shimadzu, Kyoto, Japan); the frequency range was from 4000 to 800 cm^−1^ with a resolution of 4 cm^−1^ (the number of scans was 250).

The sample for examination was a mixture of gelatin and KBr with a mass ratio of 1:220. The mixture was dissolved in distilled water then dried in a freeze dryer at −53 °C and a residual pressure of 2.4–2.6 Pa for 8–10 h [31]. The dried mixture was additionally kept in a furnace at 70 ± 5 °C for 6 h. Then, the dried mixture was compressed to form a tablet. FTIR spectra were obtained immediately after compression. 

The FTIR spectrum in the absorption region of the Amide I band (1600–1700 cm^−1^) was processed using the OriginPro 9.0. Analysis of the secondary structure of gelatin was done using the second derivative. The Amide I was decomposed into several components using a Gaussian distribution [32,33]. The quantitative contribution of each conformation (component) of the secondary structure was determined as the ratio of the integrated intensity of the corresponding band to the total integrated intensity of the amide I before decomposition.

### 2.8. Thermal Stability and Viscoelastic Properties of Gels

To form gels, the stock solutions of gelatin (C_G_ = 10 wt.%) were placed into glass beakers and stored at 5 ± 1 °C for 16–18 h before measurements.

The rheological properties of the gels were measured at shear deformation [34] using a Physica MCR301 (Anton Paar, Graz, Austrian) rheometer using a cone-and-plate unit; the diameter of the cone was 50 mm, and the angle between the cone and plate was 1 grad. 

Measurements were made in the following deformation modes:

Periodic oscillations at a constant temperature (6 °C) with varying frequency, ω, within the linear viscoelastic range at a constant amplitude of deformation γ=2%; the range of ω was 0.1–200 s^−1^; the elastic modulus, G′, and loss modulus, G″, were obtained; before frequency scanning, the amplitude of deformation scanning tests was conducted from 1% to 31% to confirm that the applied amplitude of the frequency scanning was within the linear viscoelastic range.

Temperature scanning of the samples from 5 to 20 °C (for *T*_m_) followed by 20 to 0 °C (for *T*_g_) at a scan rate of 1 °C/min, γ = 2% and ω = 6.28 c^−1^ within the linear viscoelastic range; the melting temperature, *T*_m_, and gelling temperature, *T*_g_, were calculated at temperatures where the crossover points of G′ and G″ were observed (Appendix A).

### 2.9. Differential Scanning Calorimetry (DSC)

Thermal properties of gelatin were determined using a Q2000 differential scanning calorimeter (TA Instruments, New Castle, DE, USA). The sample pan was loaded with 2–8 mg of sample and sealed hermetically. Experiments were conducted as a single run of a single sample in triplicate using the following procedure. First, the sample was cooled down to −40 °C at 20 °C/min and equilibrated at −40 °C for 5 min before the scan. Then, the sample was heated up to 150 °C with a heating rate of 10 °C/min and held at 150 °C for 5 min. Nitrogen was used as the purge gas at a continuous flow rate of 50 mL/min. Thermograms were analysed using TA Universal Analysis 2000. The melting temperature (*T*^G^_m_) of gelatin was determined as the onset of the endothermic peak observed in the first heating scan. The melting enthalpy (Δ*H*_m_) was calculated from the area under the corresponding endothermic peak. The glass transition temperature (*T*^G^_glass_) was obtained from the second heating scan and defined as the midpoint of the change in heat capacity.

### 2.10. Statistical Analysis

Experiments were done in triplicate and the data were expressed as means ± standard deviation. The data obtained were subjected to one-way ANOVA using Origin Pro 9.0. Differences among means were considered significant at *p* < 0.05.

## 3. Results

### 3.1. Chemical Composition

The chemical composition of the gelatin samples obtained from cod skin at different pH is presented in Table 1. 

The protein content in the samples ranged from 86.5% to 92.8%. The cod gelatin obtained under mild conditions at pH values of 5 and 8 (G5 and G8) was characterized by a higher protein content of 92.8% and 90.8%, respectively. In the gelatin obtained under more aggressive conditions at low pH (G3 and G4) the protein content decreased to 86–87%. A similar pattern was found for gelatin from the skin of salmon [23]. The low amino nitrogen content indicates a high quality of the extracted gelatin. It is possible that in a strongly acidic environment, not only destruction of collagen (destruction of covalent cross-links between three α-chains) occurs but also chemical hydrolysis of α-chains takes place when peptide bonds break, leading to the appearance of low molecular weight peptide fragments. This was confirmed by the higher content of amino nitrogen in G3 and G4 (Table 1). 

The main component of samples of cod gelatin is protein. Moisture, fat and mineral content affect the quality of the product and determine consumer properties considerably. No samples of cod gelatin contained fats, and the moisture content did not exceed 9 wt.% (Table 1), which is acceptable for commercially used gelatins [21]. 

The content of mineral substances in G5 was 0.6 wt.%. As the pH of extraction increased or decreased, the amount of ash increased three to four times to 2.3–2.6 wt.%. Apparently, this is due to the formation of mineral salts at the stage of neutralization during gelatin extraction from cod skin. In addition, the amount of mineral substances significantly exceeded this value in G3 compared with other gelatins. Probably, this was due not only to the neutralization stage but also to the high mineral content in the raw materials (fragments of fish scales), which are easier to dissolve in a highly acidic environment. Analysis of the data from Table 1 showed that G5 was similar in chemical composition to the control sample “Sigma”. The yield was 49–55 wt.%; all the samples were in the form of white or light-yellow powder.

### 3.2. Isoelectric Point of Gelatin

Figure 1 shows values of the isoelectric point of cod gelatin depending on the conditions (pH) of extraction. For gelatin, extracted in an acidic or alkaline medium, the values of pI were in the alkaline (from 7.1 to 9.6) or acidic (4.5 to 5.1) region, respectively. pI of gelatin “Sigma” was 7.4 (the turbidimetric method) and 7.8 (the viscometric method). 

### 3.3. Amino Acid Composition

Amino acid composition, which mainly determines the functional properties of gelatin, largely depends on the composition of raw materials. The amino acid composition of gelatin extracted from cod skin at different pH is presented in Table 2.

All samples, regardless of the extraction conditions, were characterized by a high content of glycine, proline, alanine and glutamic acid. A characteristic feature of the amino acid composition of collagen and gelatin is the high content of glycine and proline [6,16]. These amino acids form a repeating sequence: glycine–proline–X, where X is another amino acid. Such a sequence translates into the helix structure of gelatin macromolecules (at low temperatures). However, there was less glycine in G8 and G9 (~15.0 g/100 g protein) than in G3 and G4 (~20.0 g/100 g protein). Probably, this was caused by hydrolysis of the α-chains of gelatin when the cod skin was treated with an alkaline solution. The gelatin from Sigma-Aldrich (control sample ‘Sigma’) was similar to G5 in amino acid composition.

In addition, it has been determined (Table 2) that G8 and G9, obtained by alkaline extraction at pH > 7, contained a greater amount of hydroxyproline (~12.0–14.0 g/100 g protein) compared with samples obtained by acid extraction at pH < 7 (~6.9–7.5 g/100 g protein). The same result was obtained for salmon gelatin [23]. It is believed that hydroxyproline plays a key role in the stabilization of the collagen-like triple helix [12,36]. The content of other amino acids is almost independent of pH. 

### 3.4. Molecular Weight Distribution

Molecular weight distribution is an important indicator of the quality of gelatin, determining its physicochemical and functional properties [15,18]. Figure 2 shows chromatograms characterizing the molecular weight distribution of gelatin obtained at different pH. It can be seen that fish gelatin contains several fractions with different molecular weights.

Regardless of the extraction conditions, there are two peaks in the chromatograms (Figure 2): the first refers to the protein fractions with a molecular weight of about 140–150 kDa, the second to those with values of 95–110 kDa. Still, the main contribution to the value of the average molecular weight (*M*_w_) is made by the first fraction (Table 3). For gelatin obtained under more aggressive conditions (G3, G4, G8, and G9), there is a slight shift of the peaks towards lower M_w_ compared to G5. In addition, the form of the chromatogram of G5 suggests the presence of another fraction with *M*_w_ ~180 kDa. Samples obtained in an acidic medium (G3 and G4) have a wider molecular weight distribution in the low *M*_w_ range (the second peak on the chromatogram is asymmetric and has a ‘tail’, indicating molecules with a molecular weight less than 80 kDa). A wide molecular weight distribution is likely due to the hydrolytic fractionation of gelatin α-chains. According to HPLC (Figure 2), fish gelatin “Sigma” is characterized by a rather narrow molecular weight distribution and it has one fraction with *M*_w_ ~ 130 kDa.

Similar data were obtained by the PAGE method (Figure 3). Analysis of electropherograms shows the presence of two major protein fractions in cod gelatin, regardless of extraction pH. The first fraction had a molecular weight of ~110 kDa and refers to α-chains. The second fraction is characterized by a molecular weight of ~150 kDa, which suggests the presence of partially hydrolysed β-chains. The presence of α- and β-fractions is characteristic of fish gelatin [37]. The majority of fish gelatins, for example, ones from the skin of rainbow trout [16] or seabass [38] are characterized by the presence of α-chains with a molecular weight in the range of 100–120 kDa.

Thus, gelatins obtained from cod skin are characterized by a wide molecular weight distribution and an average molecular weight of ~130–150 kDa.

### 3.5. Secondary Structure of Macromolecules

The infrared spectrum of gelatin as a protein is characterized by the presence of several major absorption bands corresponding to vibrational transitions in the peptide chain (Table 4) [39,40]. Figure 4 shows FTIR spectra of cod gelatins obtained at different pH.

Analysis of the FTIR spectra showed that a change in the extraction conditions (pH) does not result in shifting of peaks. The FTIR spectra obtained are typical for fish gelatins and comparable to the results obtained for gelatins from other fish species [16,24,38].

The Amide I band (Table 4) is most sensitive to changes in the secondary structure. The complex contour of the Amide I is qualitatively explained by the superposition of bands corresponding to different conformational states of the polypeptide chain. To obtain information about the secondary structure of the protein, the FTIR spectra of gelatin were analysed using the second derivative method in the absorption region of the Amide I (see Figure 5).

Bands in the Amide I absorption region (Figure 5) were correlated with different conformational conditions of gelatin using data from reference literature [32,33,41]. The Amide I was resolved into six components which are clearly seen on the second derivative spectra (Figure 5). According to the assignments given in the literature [33,42], the main components at 1625 ± 1 and 1638 ± 1 cm^−1^ are attributed to hydrated imides with some contribution from β-sheets; the component at 1669±1 cm^−1^ results from the absorbance of collagen-like triple helices; the component centred at 1654 ± 1 cm^−1^ is attributed to a random coil; the component at 1683 ± 1 cm^−1^ is defined by the presence of β-turns; and the component at 1695 ± 2 cm^−1^ is assigned to β-sheets with some contribution from β-turn absorbance (Table 5).

The integral intensity of the corresponding peaks was determined to obtain quantitative information on the proportion of the macromolecular chain of gelatin in one or other conformation. Amide I decomposition was done using Gaussian distribution (Appendix A). Appendix A shows an example of the graphic decomposition of the amide I band into the corresponding theoretically obtained Gaussian curves for G5. Table 5 presents the results of the theoretical decomposition of the Amide I into Gaussian curves, corresponding to different conformational states of the cod gelatin macrochain obtained at different pH. 

Analysis of the data in Table 5 shows that the gelatins obtained in more aggressive conditions (G3 and G9) are characterized by a higher random coil content and lower triple helix content. It should be noted that in an acidic environment the destruction of the helices is more intensive, which is indicated by the change in triple helix content from 24.8% (G5) to 14.7% (G3). In an alkaline environment, the content of helical areas decreased from 24.8% (G5) to 19.9% (G9).

These data correlate with the results of the molecular weight distribution of cod gelatin (Figure 2 and Figure 3). Thus, samples with a high triple helix content are characterized by higher molecular weights. A similar pattern has been obtained in the study of fish gelatin from salmon [23]. 

### 3.6. Gelation and Melting Temperatures, and Viscoelastic Properties of Gels

In aqueous systems, gelatin is capable of forming hydrogels that are thermally reversible. As the temperature rises, the gels melt, i.e., gel-to-sol transition occurs. However, as the temperature is lowered, thermo-reversible sol-to-gel transition occurs. The gelling, *T*_g_, and melting, *T*_m_, temperatures of the gel are crucial characteristics of gelatin that determine its application in technologies. *T*_g_ and *T*_m_ are shown in Figure 6. For the gelatin “Sigma”, *T*_g_ = 1.5 °C and *T*_m_ = 10.7 °C.

Gelatin obtained under mild conditions (G4 and G5) formed gels with a higher melting temperature, *T*_m_~16–17 °C. As extraction pH decreased and/or increased, the gels became less thermally stable, as evidenced by a decrease in the melting temperature (Figure 6). *T*_m_ is related to the energy required to break the cross-linked junction zones [43]. Therefore, the increase in the thermostability of gels may be due either to the formation of stronger junction zones or to an increase in the number of junction zones. Consistent with our results for the secondary structure of gelatin (see Section 3.5), it would be logical to suppose that the high triple helix content in G4 and G5 is responsible for this effect.

The viscoelastic properties of the cod gelatin gels are shown in Figure 7. Strain amplitude dependencies of the storage G′ and loss G″ moduli (Figure 7a) show that the region of linear viscoelasticity is up to an amplitude of 21–26%. It should be noted that the limiting value of amplitude, γ_L_ (i.e. the boundary between linear and nonlinear regions), in some way characterizing the strength of the gel, is higher for G5 obtained under mild conditions. The constant G′ values in a wide frequency range as well as relatively low G″ values measured in the range of linear viscoelasticity (Figure 7b) indicate the solid-like type of mechanical behaviour of physical gelatin gels. One can see that variation in the extraction pH leads to a change in the storage modulus (“rigidity”) of gelatin gels.

To quantitatively evaluate the influence of extraction properties on the mechanical properties of cod gelatin gels, correlated parameters of storage modulus plateau, G′_pl_, were used (Figure 8). Gels showed an increase in G′_pl_ with a decreasing ΔpH = ∣7 − pH_extraction_∣. For example, G′_pl_ values for G5 and G8 were higher than those for G3 and G9.

The considerably lower gelling and melting temperature and storage modulus of fish gelatin gels obtained than those of mammalian gelatin [37] are probably due the lower content of proline and hydroxyproline in fish gelatin. These amino acids are of importance for the formation of some ordered structures and stabilization of the gelatin gel network [44]. Therefore, cod gelatin with a high amino acids content (Table 2) gives gels with higher gelling and melting temperatures (Figure 6) and higher storage modulus (Figure 8), confirming the formation of a network with a rigid structure. Gelatins from other marine sources: squid [45]; megrim, hake [20]; giant catfish and tilapia [46]; megrim and tuna [9] demonstrate similar results.

It is known that the gelling temperature, melting temperature and gel strength depend on the molecular weight and molecular weight distribution, that is, the ratio of α-, β- and γ-chains [47]. The fact that gelatin with higher molecular weight gives the stronger gel is well shown by experimental data (Table 3, Figure 8). The low values of *T*_m_ and G′ for gelatin gel obtained under aggressive conditions at pH 3 (G3) are explained by the presence of low molecular weight protein fractions [23,36], low triple helix content (Table 5) and, as a result, the formation of fewer (or less robust) joint zones in the gel network.

### 3.7. Melting Temperature and Glass Transition Temperature

Figure 9 presents the DSC results for gelatins obtained at different pH. The calculated values of melting temperature (*T*^G^_m1_ and *T*^G^_m2_) and glass transition temperature (*T*^G^_glass_) characterizing the thermal properties of dry gelatins are presented in Table 6. 

The DSC curves have two broad asymmetrical endothermic peaks (Figure 9a). Each peak on the heating curve corresponds to a helix–coil transition, and the number of peaks indicates the presence of several fractions. These results are consistent with the molecular weight distribution data, which also confirmed the presence of two fractions in the gelatin samples obtained (see Figure 2 and Figure 3). It can be assumed that the first peak corresponds to the melting temperature of the low molecular weight fraction (*T*^G^_m1_) with *M*_w_ = 95–110 kDa, and the second peak corresponds to the melting temperature of the high molecular weight fraction (*T*^G^_m2_) with *M*_w_ = 140–150 kDa. It is seen (Table 6) that G5 with *M*_w_ = 153 kDa, obtained under mild conditions, is characterized by a higher melting temperature (*T*^G^_m1_ = 63.7 °C and *T*^G^_m2_ = 98.7 °C) and glass transition temperature (*T*^G^_glass_ = 64.8 °C). A further shift of extraction pH to an alkaline or acidic region, i.e., a transition to more severe processing regimes, leads to a decrease in the number of triple helices (Table 5) and, as a consequence, a decrease in the weight-average molecular weight. Thus, a regular decrease in the values of *T*^G^_m_ and *T*^G^_glass_ occurs (Table 6).

In addition, the melting enthalpy (Δ*H*_m_) of the gelatins extracted under more severe conditions (G4, G8, and G9) is less than that of G5. Perhaps this is also related to the number of triple helices in the gelatin samples since the enthalpy of melting is directly proportional to their number [48]. These data are consistent with the results of molecular weight distribution and FTIR spectroscopy. Thus, G5 with Δ*H*_m_ = 24.1 J/g (Table 6) is characterized by a higher triple helix content: 24.8% (Table 5) and higher molecular weight: 153 kDa (Table 3). Similar patterns were not found in G3 for which the maximum values of *T*^G^_m2_ and Δ*H*_m_ were, respectively, 118.6 °C and 63.5 J/g. Perhaps this is due to the increased mineral content in this sample (Table 1).

Thus, G4, G8, and G9, obtained under more aggressive conditions, are characterized by a lower triple helix content and, consequently, by low values of the melting enthalpy compared with G5.

## 4. Conclusions

Gelatin from the skin of Atlantic cod extracted at various pH values of the aqueous phase (in the range from pH 3 to 9) has different characteristics and functional properties. Gelatin obtained under mild conditions (G5) is characterized by a higher protein content. However, all samples of gelatin are characterized by a fairly high protein content and a lack of fat. The amount of moisture and mineral substances in the samples does not exceed the acceptable value for commercial gelatins. The yield of gelatin was 49–55 wt.%. The extraction pH significantly affects the amino acid composition, the molecular weight distribution and the secondary structure (the number of triple helices) of cod skin gelatin macromolecules. Gelatin G5, extracted in a weak acidic environment, is characterized by a higher molecular weight and a high content of ordered structures, i.e., collagen-like triple helices and amino acids (proline and hydroxyproline). As a result, this sample of gelatin has a high thermal stability of the polymer chains compared to the samples obtained at lower pH or in an alkaline environment. This determines the functional properties of gelatin, namely the high temperature of gel formation and melting, and a high elasticity modulus of the gel. Consequently, the skin of Atlantic cod could be used as a raw material for gelatin extraction when the extraction is conducted at appropriate pH.

## Figures and Tables

**Figure 1 polymers-11-01724-f001:**
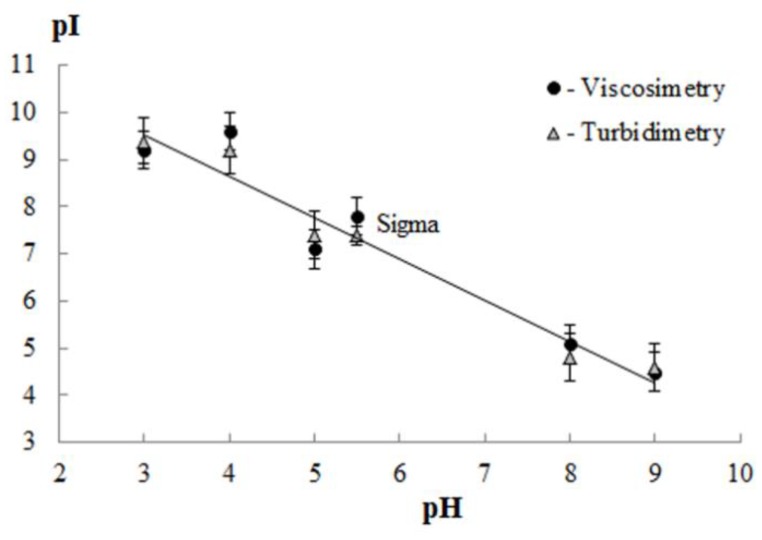
Effect of extraction pH on the isoelectric point (pI) of gelatin. Data were obtained by the methods of turbidimetry and capillary viscometry.

**Figure 2 polymers-11-01724-f002:**
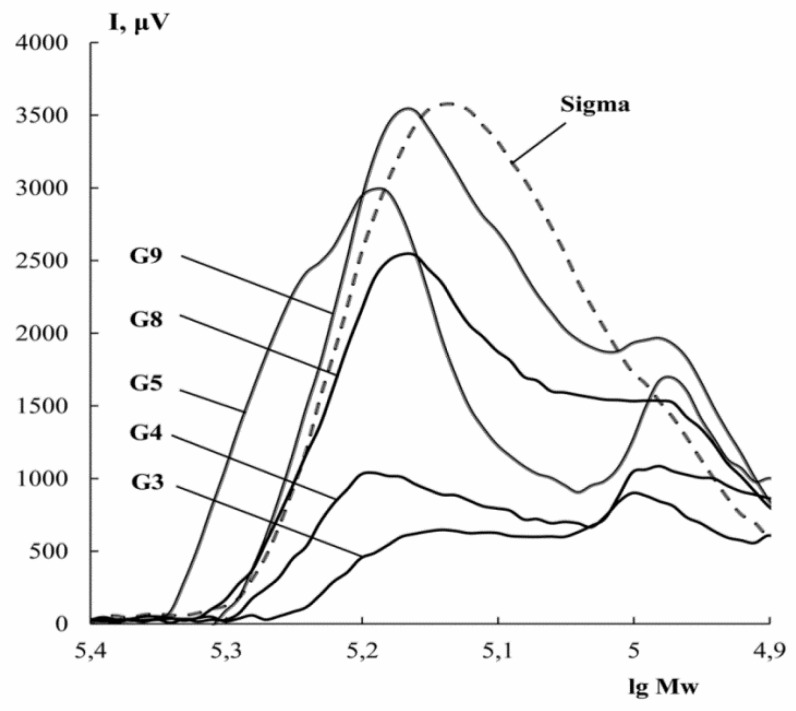
The molecular weight distribution of cod gelatin samples obtained at different pH. Extraction pH is shown as numbers with lines pointing towards curves. Chromatograms were obtained by an high-performance liquid chromatography (HPLC) method.

**Figure 3 polymers-11-01724-f003:**
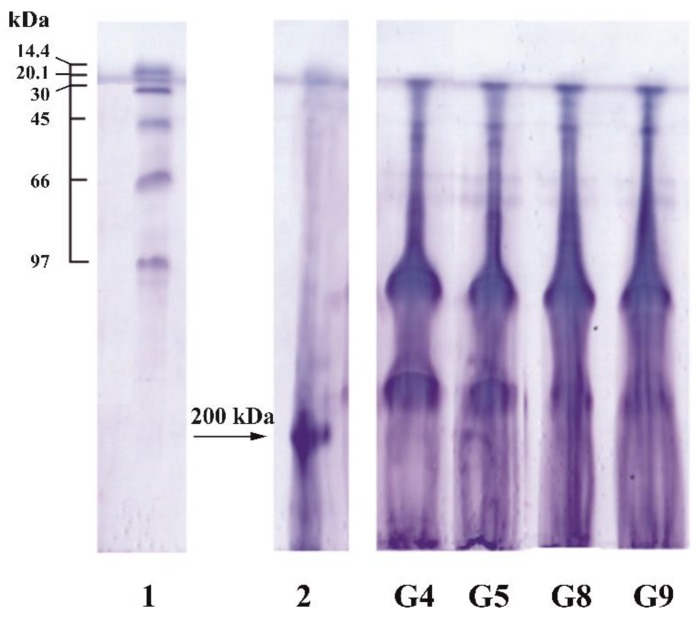
Electropherogram of gelatin samples obtained from cod skin at different pH. Electropherograms of standard low molecular weight markers (GE Healthcare) (1) and a reference sample with *M*_w_ = 200 kDa (2) are shown.

**Figure 4 polymers-11-01724-f004:**
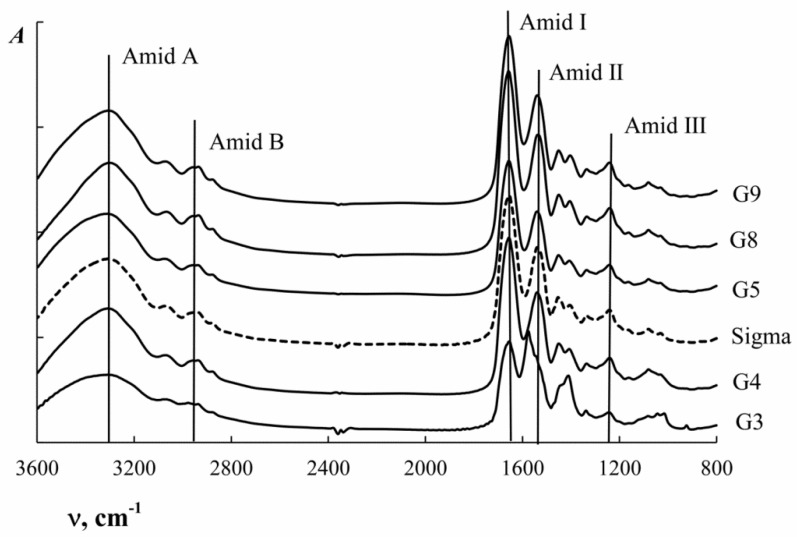
FTIR absorption spectra of cod gelatin samples obtained at different pH. pH values are given next to the appropriate spectra.

**Figure 5 polymers-11-01724-f005:**
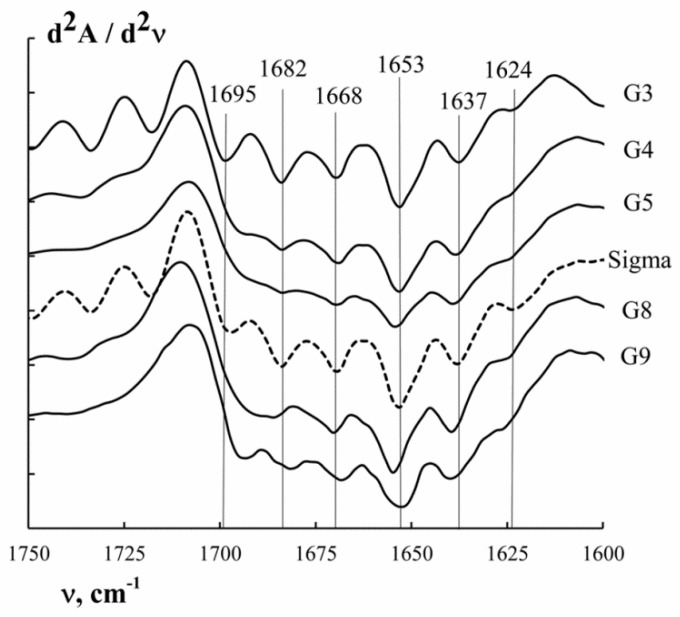
Spectra of the second derivative (differential FTIR spectra) in the absorption region of amide I.

**Figure 6 polymers-11-01724-f006:**
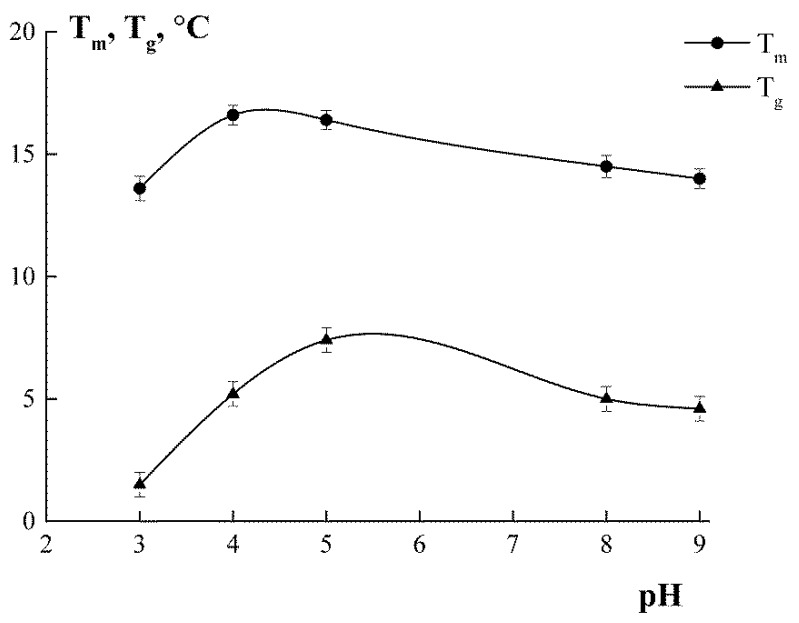
Gelling, *T*_g_, and melting, *T*_m_, temperatures of cod gelatin gels depending on extraction pH. C_G_ = 10 wt.%.

**Figure 7 polymers-11-01724-f007:**
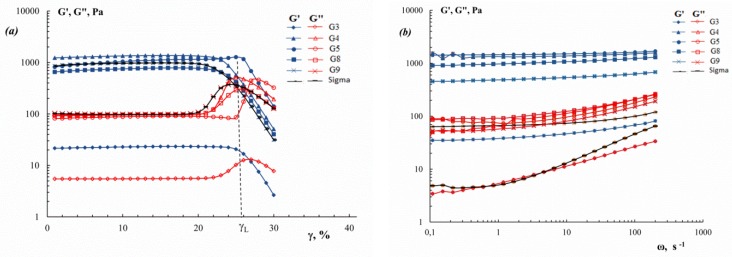
(**a**) Strain amplitude dependencies of the storage modulus G′ (filled points) and loss modulus G″ (open points), ω = 6.28 s^−1^; (**b**) Frequency dependence of storage modulus G′ (filled points) and loss modulus G″ (open points), γ = 2%, for gels of cod gelatin extracted at different pH. C_G_ = 10 wt.%, t = 6 °C.

**Figure 8 polymers-11-01724-f008:**
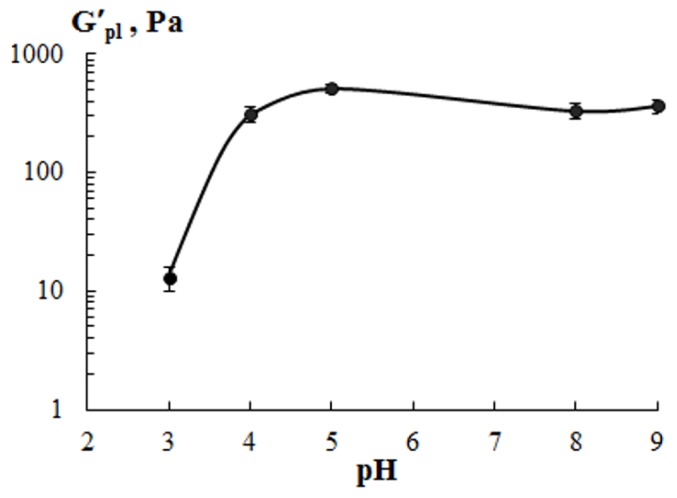
Effect of extraction conditions on the storage modulus plateau (G′_pl_) for cod gelatin gels. C_G_ = 10 wt.%.

**Figure 9 polymers-11-01724-f009:**
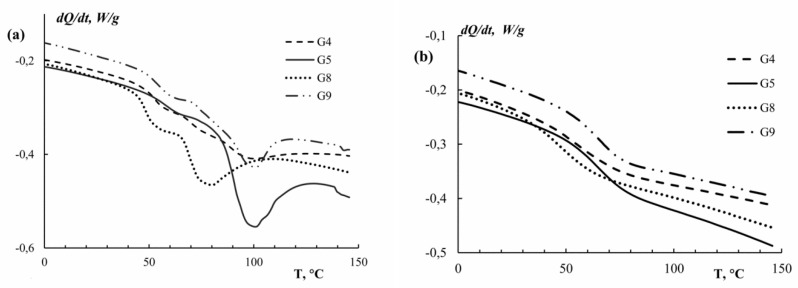
Thermograms of gelatin samples obtained at different pH (**a**) First heating cycle—determination of melting temperature and enthalpy; (**b**) second heating cycle—determination of glass transition temperature; the heating rate of 10 °C/min.

**Table 1 polymers-11-01724-t001:** Chemical composition of gelatin samples obtained from cod skin at different pH values of extraction.

Sample	Extraction pH	Moisture Content X, %	Amine Nitrogen N_A_, %	Total Nitrogen N_T_, %	Protein * P, %	Ash, %	Yield B, %
G3	3.0	8.5 ± 0.5 ^d^	0.96 ± 0.03 ^d^	15.6 ± 0.1 ^a^	86.5 ± 0.6 ^a^	4.8 ± 0.2 ^d^	51.1 ± 0.6 ^a^
G4	4.0	9.0 ± 0.7 ^c^	1.04 ± 0.04 ^c^	15.8 ± 0.2 ^b^	87.7 ± 1.0 ^b^	2.6 ± 0.2 ^c^	51.2 ± 0.5 ^a^
G5	5.0	6.0 ± 0.5 ^a^	0.76 ± 0.05 ^b^	16.8 ± 0.1 ^c^	92.8 ± 0.6 ^c^	0.6 ± 0.1 ^b^	55.4 ± 0.3 ^b^
G8	8.0	7.0 ± 0.5 ^b^	0.70 ± 0.07 ^a^	16.5 ± 0.1 ^d^	90.8 ± 0.6 ^d^	2.3 ± 0.1 ^a^	49.3 ± 0.4 ^c^
G9	9.0	7.5 ± 0.4 ^b^	0.69 ± 0.08 ^a^	16.2 ± 0.1 ^e^	89.8 ± 0.6 ^e^	2.4 ± 0.1 ^a^	49.1 ± 0.4 ^c^
Sigma **	-	5.5 ± 0.5 ^a^	0.53 ± 0.03 ^e^	17.0 ± 0.1 ^c^	94.4 ± 0.6 ^c^	-	

* Mass fraction of protein was calculated as *P* = N_T_ × 5.55 (5.55 is the conversion coefficient from nitrogen to collagen [35]). ** Control sample. Date are means ± standard deviation (n = 3). Different superscript letters in the same column indicate significant differences (*P* < 0.05).

**Table 2 polymers-11-01724-t002:** Amino acid composition (amino acid content, g/100 g protein) of cod gelatin samples obtained at different pH and of a control sample.

Amino Acids	G3	G4	G5	G8	G9	‘Sigma’
Glycine	20.5	20.2	18.5	15.2	14.9	18.6
Proline	11.8	13.0	12.2	13.8	13.9	12.9
Hydroxyproline	6.9	7.2	7.5	14.6	11.5	9.6
Aspartic acid	6.2	6.1	5.6	6.2	6.2	5.6
Glutamic acid	9.5	9.4	9.1	9.4	9.6	9.3
Serine	6.7	6.7	6.6	6.3	6.6	6.3
Histidine	1.6	1.6	1.9	1.3	1.5	1.7
Threonine	2.6	2.6	2.7	2.6	2.7	2.6
Arginine	7.7	7.9	7.7	7.6	7.9	7.6
Alanine	9.5	9.1	9.3	8.5	9.0	9.4
Taurine	3.4	3.1	3.7	1.8	2.5	2.9
Tyrosine	0.9	0.7	1.0	0.8	0.9	0.8
Valine	2.0	2.0	2.1	2.1	2.1	2.1
Methionine	1.4	1.5	1.8	1.3	1.4	1.6
Isoleucine	1.4	1.3	1.6	1.4	1.6	1.5
Leucine	2.8	2.7	2.9	2.6	2.8	2.8
Lysine	2.8	2.9	3.5	2.4	2.6	2.3
Phenylalanine	2.3	2.0	2.3	2.1	2.3	2.4

**Table 3 polymers-11-01724-t003:** Molecular weight distribution of cod gelatin according to the HPLC method.

Samples	*M*_wf_, kDa	ω, % *	*M*_W_, kDa **
G3	≤ 80	13.3 ± 2.8	115.2 ± 6.8 ^c^
	96.2 ± 5.5 ^a^	38.2 ± 3.5 ^a^	
	139.3 ± 6.2 ^b^	48.5 ± 3.1 ^d^	
G4	103.5 ± 4.8 ^a^	42.1 ± 2.4 ^a^	131.6 ± 7.2 ^a^
	152.1 ± 6.1 ^b^	57.9 ± 3.9 ^b^	
G5	112.0 ± 6.9 ^a^	22.4 ±2.2 ^c^	153.0 ± 9.0 ^b^
	151.0 ± 6.8 ^b^	39.4 ± 3.1 ^a^	
	179.1 ± 7.0 ^c^	38.2 ± 3.6 ^a^	
G8	105.0 ± 6.0 ^a^	41.7 ± 2.8 ^a^	130.2 ± 8.3 ^a^
	148.3 ± 7.9 ^b^	58.3 ± 4.0 ^b^	
G9	100.5 ±6.1 ^a^	35.6 ± 3.1 ^a^	129.2 ± 8.5 ^a^
	143.9 ± 7.8 ^b^	64.4 ± 4.2 ^b^	

*M*_wf_, kDa—molecular weight of fraction; * ω, %—proportion of each fraction (determined using the Gaussian distribution); ** *M*_W_, kDa—calculated by the additivity rule. Date are means ± standard deviation (n = 3). Different superscript letters in the same column indicate significant differences (*P* < 0.05).

**Table 4 polymers-11-01724-t004:** Main absorption bands of the functional groups of gelatin [39,40].

Group	Wave Number ν, cm^−1^	Type of Vibration
Amide A	3400–3300	N–H stretching vibrations
Amide B	3000–2900	N–H stretching vibrations
Amide I	1700–1600	C=O stretching vibrations—80% and C–N stretching vibrations
Amide II	1575–1480	N–H deformation vibrations—80% and C–N stretching vibrations
Amide III	1300–1230	C–N stretching vibrations

**Table 5 polymers-11-01724-t005:** Content (as a percentage) of major components of the secondary structure of the protein in cod gelatin samples obtained at different pH.

Secondary Structure Elements	Wave Number ν, cm^−1^	G3	G4	G5	G8	G9	“Sigma”
β-Turn/β-sheet	1624–1626	12.8	10.5	13.6	11.9	12.4	13.5
	1637–1639	14.9	18.5	16.1	16.3	16.3	14.0
Random coil	1653–1655	23.8	20.3	20.1	18.6	23.1	24.9
Triple helix	1668–1670	14.7	19.7	24.8	26.9	19.9	13.8
β-Turn/β-sheet	1682–1684	24.4	28.4	20.5	18.9	24.1	29.0
	1692–1697	9.4	2.6	4.9	7.4	4.2	4.8

**Table 6 polymers-11-01724-t006:** Thermal properties of cod gelatin obtained at different pH.

Sample	*T*^G^_m1_, °C	*T*^G^_m2_, °C	Δ*H*_m_, J/g	*T*^G^_glass_, °C
G3	56.4 ± 0.8 ^a^	118.6 ± 1.2 ^e^	63.5 ± 1.0 ^a^	45.4 ± 0.7 ^f^
G4	56.7 ± 0.8 ^a^	97.3 ± 0.7 ^d^	12.9 ± 0.8 ^b^	56.8 ± 0.6 ^e^
G5	63.7 ± 0.7 ^b^	98.7 ± 0.9 ^c^	24.1 ± 0.7 ^c^	64.8 ± 0.9 ^d^
G8	51.7 ± 0.5 ^c^	78.7 ± 0.6 ^b^	24.0 ± 0.8 ^c^	46.8 ± 0.5 ^c^
G9	59.4 ± 0.6 ^d^	82.7 ± 0.7 ^a^	15.8 ± 0.6 ^d^	65.5 ± 0.7 ^b^
Sigma *	74.3 ± 0.5		13.3 ± 0.4 ^e^	61.7 ± 0.4 ^a^

* Control sample. Date are means ± standard deviation (n = 3). Different superscript letters in the same column indicate significant differences (*P* < 0.05).

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
