# Peer review of "Tailoring Cod Gelatin Structure and Physical Properties with Acid and Alkaline Extraction"

_polymers, 2019, doi:10.3390/polym11101724_

Round 1
Reviewer 1 Report
Introduction. The antecedents are well exposed and referenced. The main topic of the article is clearly exposed. However, a greater depth in the influence of pH on the quality of gelatin, the main topic to be investigated in the article, is missing.
Materials and methods. The section of materials and methods is well explained and includes the necessary references in order to understand the methodology used. However, the authors should clarify the following points:
Where did the used fish come from? What was the procedure for defatting the skins? If the authors consider that the methodology has already been explained above, cite bibliography in this regard.Results. The results provided are presented clearly and comprehensively. The results are significant and make it feasible to achieve the objectives set by the authors. However, some aspects should be clarified:
- Although all the samples obtained have a humidity of less than 10%, what is the difference in humidity between them due to?
- The content of amine nitrogen and total nitrogen is analyzed, however it is not mentioned in the discussion of the results in Table 1.
- Has the molecular weight of the reference gelatin (sigma) in Table 3 not been evaluated? It would be interesting to know if it maintains a molecular weight similar to the G5 sample.
Conclusions
The conclusions are presented and reflect the results obtained.
Author Response
Please, see the attachment.

Reviewer 2 Report
This manuscript focus on the properties of gelatin that was extracted from the skin of Atlantic cod at different pH, and find that at weak acid condition (pH5), the gelatin had a higher molecular weight, higher amounts of ordered triple collagen-like helices, higher glass transition temperature and melting enthalpy, and higher extraction yield and protein content. This work can be accepted; however care should be taken for some issues as given below. - In tables 2 and 3, the standard deviations were not provided. - As mentioned in the manuscript, "gelatin from cold-water fish has inferior thermal and rheological properties of gelatin to that extracted from warm-water fish." Authors should tell what is the potential applications or advantages of gelatin from cold-water fish (In what circumstance, we need gelation with lower melting point or weak rheological properties; or else, is there any way to enhance these properties by the post treatment of the gelatin for applications).Author Response
Please see the attachment.
